# Bridging the Gap between Sample-based and One-shot Neural Architecture Search with BONAS

**Han Shi**[1]*, **Renjie Pi**[2]*, **Hang Xu**[3], **Zhenguo Li**[3], **James T. Kwok**[1], **Tong Zhang**[1]
[1]Hong Kong University of Science and Technology, Hong Kong
{hshiac,jamesk}@cse.ust.hk, tongzhang@ust.hk
[2]The University of Hong Kong, Hong Kong
pipilu@hku.hk
[3]Huawei Noah's Ark Lab
{xu.hang,li.zhenguo}@huawei.com

## Abstract

Neural Architecture Search (NAS) has shown great potentials in finding better neural network designs. Sample-based NAS is the most reliable approach which aims at exploring the search space and evaluating the most promising architectures. However, it is computationally very costly. As a remedy, the one-shot approach has emerged as a popular technique for accelerating NAS using weight-sharing. However, due to the weight-sharing of vastly different networks, the one-shot approach is less reliable than the sample-based approach. In this work, we propose BONAS (Bayesian Optimized Neural Architecture Search), a sample-based NAS framework which is accelerated using weight-sharing to evaluate multiple related architectures simultaneously. Specifically, we apply a Graph Convolutional Network predictor as surrogate model for Bayesian Optimization to select multiple related candidate models in each iteration. We then apply weight-sharing to train multiple candidate models simultaneously. This approach not only accelerates the traditional sample-based approach significantly, but also keeps its reliability. This is because weight-sharing among related architectures is more reliable than that in the one-shot approach. Extensive experiments are conducted to verify the effectiveness of our method over competing algorithms.[1]

## 1 Introduction

Designing an appropriate deep network architecture for each task and data set is tedious and time-consuming. Neural architecture search (NAS) [42], which attempts to find this architecture automatically, has aroused significant interest recently. Results competitive with hand-crafted architectures have been obtained in many application areas, such as natural language processing [20, 30] and computer vision [25, 9, 3, 16, 4].

Optimization in NAS is difficult because the search space can contain billions of network architectures. Moreover, the performance (e.g., accuracy) of a particular architecture is computationally expensive to evaluate. Hence, a central component in NAS is the strategy to search such a huge space of architectures. These strategies can be broadly categorized into two groups. Sample-based algorithms [41, 17, 25, 19], which perform NAS in two phases: (i) search for candidate architectures with potentially good performance; and (ii) query their actual performance by full training. The second category contains one-shot NAS algorithms, which combine architectures in the whole search space together using weight sharing [24] or continuous relaxation [18, 19] for faster evaluation. Despite

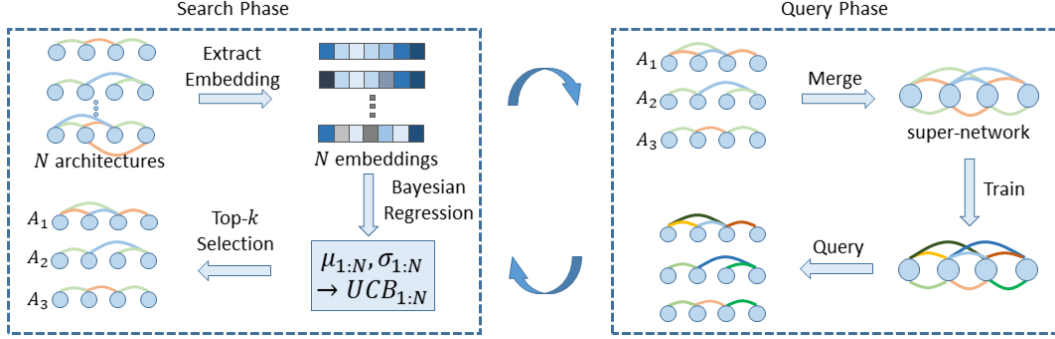

Figure 1: Overview of the proposed BONAS. In the search phase, we use GCN embedding extractor and Bayesian Sigmoid Regression as the surrogate model for Bayesian Optimization and multiple candidate architectures are selected. In the query phase, we merge them as a super-network. Based on the trained super-network, we can query each sub-network using corresponding paths.

their appealing speed, one-shot algorithms suffer from the following: (i) The obtained result can be sensitive to initialization, which hinders reproducibility; (ii) Constraints need to be imposed on the search space so as to constrain the super-network size (otherwise, it may be too large to fit in the memory). As opposed to one-shot algorithms, sample-based approaches are more flexible with respect to the search space, and can usually find promising architectures regardless of initialization. However, the heavy computation required by sample-based methods inevitably becomes the major obstacle. In this paper, we aim to develop a more efficient sample-based NAS algorithm while taking advantage of the weight-sharing paradigm.

Due to the large sizes of most search spaces, searching for competitive architectures can be very difficult. To alleviate this issue, Bayesian optimization (BO) [23], which explicitly considers exploitation and exploration, comes in handy as an efficient model for search and optimization problems. In BO, a commonly used surrogate model is the Gaussian process (GP) [28]. However, its time complexity increases cubically with the number of samples [29]. Hence, it is costly for use in NAS due to the huge search space. Another drawback of GP in NAS is that it requires a manually designed kernel on architectures. While two heuristically-designed kernels are provided in [10] and [11], they can neither be easily adapted to different architecture families nor be further optimized based on data. It is still an open issue on how to define a good neural architecture kernel. On the other hand, in the query phase, the traditional approach of fully training the neural architectures is costly. Although early stopping can be adopted [33, 35], it cannot reduce the training time substantially while inevitably compromising the fidelity of the obtained results. In one-shot methods, weight-sharing is performed on the whole space of sub-networks [24]. These sub-networks can be very different and so sharing their weights may not be a good idea.

To alleviate these problems, we present BONAS (Bayesian Optimized Neural Architecture Search), which is a sample-based NAS algorithm combined with weight-sharing (Figure 1). In the search phase, we first use a graph convolutional network (GCN) [13] to produce embeddings for the neural architectures. This naturally handles the graph structures of neural architectures, and avoids defining GP's kernel function. Together with a novel Bayesian sigmoid regressor, it replaces the GP as BO's surrogate model. In the query phase, we construct a super-network from a batch of promising candidate architectures, and train them by uniform sampling. These candidates are then queried simultaneously based on the learned weight of the super-network. As weight-sharing is now performed only on a small subset of similarly-performing sub-networks with high BO scores, this is more reasonable than sharing the weights of all sub-networks in the search space as in one-shot NAS methods [24].

Empirically, the proposed BONAS outperforms state-of-the-art methods. We observe consistent gains on multiple search spaces for vision and NLP tasks. These include the standard benchmark data sets of NAS-Bench-101 [38] and NAS-Bench-201 [8] on convolutional architectures, and a new NAS benchmark data set LSTM-12K we recently collected for LSTMs. The proposed algorithm also finds competitive models efficiently in open-domain search with the NASNet search space [42].

The contributions of this paper are as follows. (i) We improve the efficiency of sample-based NAS using Bayesian optimization in combination with a novel GCN embedding extractor and Bayesian Sigmoid Regression to select candidate architectures. (ii) We accelerate the evaluation of sample-based NAS by training multiple related architectures simultaneously using weight-sharing. (iii) Extensive experiments on both closed and open domains demonstrate the efficiency of the proposed method. BONAS achieves consistent gains on different benchmarks compared with competing baselines. It bridges the gap between training speeds of sample-based and one-shot NAS methods.

## 2 Related Work

### 2.1 Bayesian Optimization

Bayesian optimization (BO) [23], with the Gaussian process (GP) [28] as the underlying surrogate model, is a popular technique for finding the globally optimal solution of an optimization problem. To guide the search, an acquisition function is used to balance exploitation and exploration [27]. Common examples include the maximum probability of improvement (MPI) [14], expected improvement (EI) [23] and upper confidence bound (UCB) [31]. In this paper, we focus on the UCB, whose exploitation-exploration tradeoff is explicit and easy to adjust. Let the hyperparameters of BO's surrogate model be $\Theta$, and the observed data be $\mathcal{D}$. The UCB for a new sample $x$ is:

$$a_{\text{UCB}}(x; \mathcal{D}, \Theta) = \mu(x; \mathcal{D}, \Theta) + \gamma \sigma(x; \mathcal{D}, \Theta), \tag{1}$$

where $\mu(x; \mathcal{D}, \Theta)$ is the predictive mean of the output from the surrogate model, $\sigma^2(x; \mathcal{D}, \Theta)$ is the corresponding predictive variance, and $\gamma > 0$ is a tradeoff parameter. A larger $\gamma$ puts more emphasis on exploration, and vice versa.

### 2.2 BO for Neural Architecture Search

Recently, BO is also used in NAS [10, 11]. Its generic procedure is shown in Algorithm 1. Since the NAS search $\mathcal{A}$ is huge, each BO iteration typically only considers a pool of architectures, which is generated, for example, by an evolutionary algorithm (EA) [25]. An acquisition function score is computed for each architecture in the pool, and architectures with the top scores are then selected for query. The procedure is repeated until convergence.

| **Algorithm 1** Generic BO procedure for NAS. | **Algorithm 2** BONAS. |
|---|---|
| 1: randomly select $m_0$ architectures $\mathcal{D}$ from search space $\mathcal{A}$ for full training; | 1: randomly select $m_0$ architectures $\mathcal{D}$ from search space $\mathcal{A}$ for weight-sharing training; |
| 2: initialize surrogate model using $\mathcal{D}$; | 2: initialize GCN and BSR using $\mathcal{D}$; |
| 3: **repeat** | 3: **repeat** |
| 4:  sample candidate pool $\mathcal{C}$ from $\mathcal{A}$; | 4:  sample candidate pool $\mathcal{C}$ from $\mathcal{A}$ by EA; |
| 5:  **for** each candidate $m$ in $\mathcal{C}$ **do** | 5:  **for** each candidate $m$ in $\mathcal{C}$ **do** |
| 6:   score $m$ using acquisition function; | 6:   embed $m$ using GCN; |
| 7:  **end for** | 7:   compute mean and variance using BSR; |
| 8:  $M \leftarrow$ candidate(s) with the top score(s); | 8:   compute UCB in (1); |
| 9:  (query): obtain actual performance of $M$; | 9:  **end for** |
|  | 10:  $M \leftarrow$ candidates with the top-$k$ scores; |
| 10:  add $M$ and its performance to $\mathcal{D}$; | 11:  (query): train $M$ with weight-sharing; |
| 11:  update surrogate model with the enlarged $\mathcal{D}$; | 12:  add $M$ and their performances to $\mathcal{D}$; |
|  | 13:  update GCN and BSR with the enlarged $\mathcal{D}$; |
| 12: **until** convergence. | 14: **until** convergence. |

## 3 Proposed Method

As can be seen from Algorithm 1, the key issues for the successful application of BO for NAS are: (i) How to represent an architecture? (ii) How to find good candidates with the surrogate model? In particular, candidates with high acquisition scores should have high actual performance; (iii) How to query the selected candidates efficiently?

In this section, we design a surrogate model combining a GCN embedding extractor and a Bayesian sigmoid regressor (Section 3.1). To alleviate the cost of full training in each query, we adopt the weight-sharing paradigm to query a batch of promising architectures together (Section 3.2). Figure 1 shows an overview of the proposed algorithm (Section 3.3).

## 3.1 Finding Potential Candidates with the Surrogate Model

In this section, we introduce a surrogate model which consists of a GCN embedding extractor and a Bayesian sigmoid regressor.

### 3.1.1 Representing Neural Networks using GCN

A neural network can be represented as a directed attributed graph. Each node represents an operation (such as a $1 \times 1$ convolution in CNN, and ReLU activation in LSTM) while edges represent data flows [39]. Figure 2 shows an example on NAS-Bench-101. Since a NAS-Bench-101 architecture is obtained by stacking multiple repeated cells, we only consider the embedding of such a cell. Graph connectivity is encoded by the adjacency matrix $\boldsymbol{A}$. Individual operations are encoded as one-hot vectors, and then aggregated to form the feature matrix $\boldsymbol{X}$.

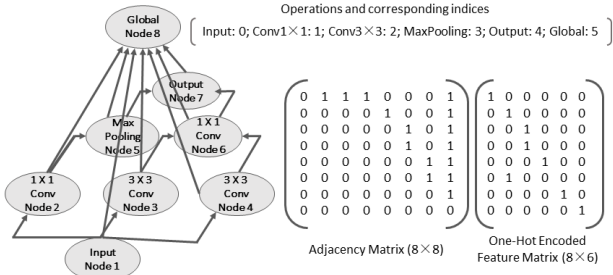

Figure 2: Encoding of an example cell in NAS-Bench-101.

Existing works often use MLP or LSTM to encode networks [17, 19, 34]. However, it is more natural to use GCN, which can well preserve the graph's structural information [13]. Besides, a MLP only allows fixed-sized inputs, while a GCN can handle input graphs with variable numbers of nodes.

The standard GCN is only used to produce node embeddings [13], while here the target is to obtain an embedding for the whole graph. To solve this problem, following [26], we connect all nodes in the graph to an additional "global" node (Figure 2). The one-hot encoding scheme is also extended to include these new connections as a new operation. The embedding of the global node is then used as the embedding of the whole graph.

To train the GCN, we feed its output (cell embedding) to a regressor for accuracy prediction. In the experiments, we use a single-hidden-layer network with sigmoid function, which constrains the prediction to be in $[0, 1]$ (it can be easily scaled to a different range when another performance metric is used). This regressor is then trained end-to-end with the GCN by minimizing the square loss.

BANANAS [35], an independent concurrent work with this paper, also tries to encode the graph structure by a so-called *path encoding* scheme, which is then fed to a MLP (called *meta neural network*) for performance estimation. The path encoding scheme is similar to the bag-of-words representation for documents, and its combined use with a simple MLP is less powerful than the GCN (as will be demonstrated empirically in Section 4.1). Moreover, its encoding vector scales exponentially in size with the number of nodes, and so may not be scalable to large cells. In [35], they need to truncate the encoding by eliminating paths that are less likely.

### 3.1.2 Bayesian Sigmoid Regression

To compute the mean and variance of the architecture's performance in (1), we introduce a Bayesian sigmoid regression (BSR) model. This is inspired by the Bayesian linear regression (BLR) in neural networks [29]. For an architecture (graph) with adjacency matrix $\boldsymbol{A}$ and feature matrix $\boldsymbol{X}$, let $\phi(\boldsymbol{A}, \boldsymbol{X})$ be the learned embedding in Section 3.1.1. Given a set $\mathcal{D}$ of $N$ trained architectures $\{(\boldsymbol{A}_i, \boldsymbol{X}_i)\}$ with known performance (accuracy) values $\{t_i\}$, the corresponding embedding vectors $\{\phi(\boldsymbol{A}_i, \boldsymbol{X}_i)\}$ are stacked to form a design matrix $\boldsymbol{\Phi}$ with $\Phi_{ij} = \phi_j(\boldsymbol{A}_i, \boldsymbol{X}_i)$. Recall from Section 3.1.1 that the final layer of the GCN predictor contains a sigmoid function (rather than the

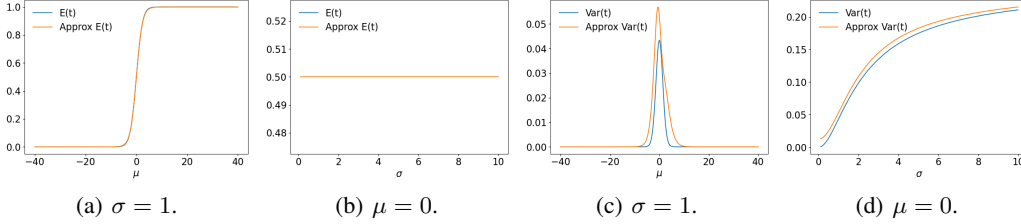

|  |  |  |  |
|---|---|---|---|
| (a) $\sigma = 1$. | (b) $\mu = 0$. | (c) $\sigma = 1$. | (d) $\mu = 0$. |

Figure 3: Plots for the true and approximate values of $E[t]$ and $var[t]$.

linear function in BLR). Instead of fitting the true performance $t$, we estimate the value before the sigmoid, i.e., $y = \text{logit}(t) = \log(t/(1-t))$, such that we can convert nonlinear regression to a linear regression problem. We denote the vector of regression values by $\boldsymbol{y} = [y_1, y_2, \dots, y_N]^T$.

In Section 3.1.1, the GCN is trained with the square loss. However, when training BO's surrogate model, we are more interested in predicting high-performance architectures as accurately as possible. Inspired by the focal loss in classification [15], we use the following exponentially weighted loss, which puts more emphasis on models with higher accuracies:

$$L_{exp} = \frac{1}{N} \sum_{i=1}^{N} (\exp(t_i) - 1)(\widetilde{t_i} - t_i)^2, \tag{2}$$

where $\widetilde{t_i}$ is GCN's predicted accuracy on architecture $i$.

For a candidate network $(\boldsymbol{A}, \boldsymbol{X})$, BO has to evaluate its acquisition score. For UCB, this involves estimating the predictive mean and predictive variance of the predicted accuracy. From [2], the predictive mean $\mu$ of $\text{logit}(t)$ is:

$$\mu(\boldsymbol{A}, \boldsymbol{X}; \mathcal{D}, \alpha, \beta) = \boldsymbol{m}_N^T \phi(\boldsymbol{A}, \boldsymbol{X}), \tag{3}$$

where $\boldsymbol{m}_N = \beta \boldsymbol{S}_N \boldsymbol{\Phi}^T \boldsymbol{y}$, $\boldsymbol{S}_N = (\alpha \boldsymbol{I} + \beta \boldsymbol{\Phi}^T \boldsymbol{\Phi})^{-1}$, $\boldsymbol{I}$ is the identity matrix, and $(\alpha, \beta)$ are precision parameters that can be estimated by maximizing the marginal likelihood [28]. By considering only the weight uncertainty in the last layer, the predictive variance of $\text{logit}(t)$ is [2]:

$$\sigma^2(\boldsymbol{A}, \boldsymbol{X}; \mathcal{D}, \alpha, \beta) = \phi(\boldsymbol{A}, \boldsymbol{X})^T \boldsymbol{S}_N \phi(\boldsymbol{A}, \boldsymbol{X}) + 1/\beta. \tag{4}$$

In the following, we show how to convert this to the predictive variance of $t$.

First, note that $t$ follows the logit-normal distribution[2] [22]. However, its $E[t]$ and $var[t]$ cannot be analytically computed. To alleviate this problem, we rewrite $E[t]$ and $E[t^2]$ as

$$E[t] = \int \text{sigmoid}(x) \mathcal{N}(x|\mu, \sigma^2) dx, \quad E[t^2] = \int (\text{sigmoid}(x))^2 \mathcal{N}(x|\mu, \sigma^2) dx,$$

where $\mathcal{N}(x|\mu, \sigma^2)$ is the normal distribution with mean $\mu$ and variance $\sigma^2$. Let $\Phi(x) = \int_{-\infty}^{x} \mathcal{N}(z|0, 1) dz$ be the cumulative distribution function of $\mathcal{N}(x|0, 1)$. We approximate $\text{sigmoid}(x)$ with $\Phi(\lambda x)$ for some $\lambda$, and similarly $(\text{sigmoid}(x))^2$ with $\Phi(\lambda \alpha(x + \beta))$, for some $\lambda, \alpha, \beta$. With these approximations, the following Proposition shows that the integrals can be analytically computed. Proof is in Appendix A.

**Proposition 1.** *For given $\alpha$ and $\beta$, $\int \Phi(\alpha(x + \beta)) \mathcal{N}(x|\mu, \sigma^2) dx = \Phi\left( \frac{\alpha(\mu + \beta)}{(1 + \alpha^2 \sigma^2)^{1/2}} \right)$.*

**Corollary 1.** *The expectation and variance of the logit-normal distribution can be approximated as:*

$$E[t] \simeq \text{sigmoid}\left( \frac{\mu}{\sqrt{1 + \lambda^2 \sigma^2}} \right), \quad var[t] \simeq \text{sigmoid}\left( \frac{\alpha(\mu + \beta)}{\sqrt{1 + \lambda^2 \alpha^2 \sigma^2}} \right) - \left( \text{sigmoid}\left( \frac{\mu}{\sqrt{1 + \lambda^2 \sigma^2}} \right) \right)^2,$$

*where $\lambda^2 = \pi/8, \alpha = 4 - 2\sqrt{2}$ and $\beta = -\log(\sqrt{2} + 1)$.*

Figure 3 shows illustrations of the approximations. From the obtained $E[t]$ and $var[t]$, one can plug into the UCB score in (1), and use this to select the next (architecture) sample from the pool.

BANANAS [35] also uses BO to search. To obtain the predictive variance in UCB, they compute the empirical variance over the outputs of an ensemble of MLP predictors. A small ensemble leads to biased estimation, while a large ensemble is computationally expensive to train and store. In contrast, we only use one GCN predictor, and the variance is obtained directly from BSR.

## 3.2 Efficient Estimation of Candidate Performance

In sample-based NAS algorithms, an architecture is selected in each iteration for full training [25, 33, 35, 42], and is computationally expensive. To alleviate this problem, we select in each BO iteration a batch of $k$ architectures $\{(\boldsymbol{A}_i, \boldsymbol{X}_i)\}_{i=1}^k$ with the top-$k$ UCB scores, and then train them together as a super-network by weight-sharing. While weight-sharing has been commonly used in one-shot NAS algorithms [18, 24, 36], their super-networks usually contain architectures from the whole search space. This makes it infeasible to train each architecture fairly [5], and some architectures may not be trained as sufficiently as others. In contrast, we only use a small number of architectures to form the super-network ($k = 100$ in the experiments). With such a small $k$, training time can be allocated to the sub-networks more evenly. Moreover, since these $k$ chosen architectures have top UCB scores, they are promising candidates and likely to contain common useful structures for the task. Thus, weight-sharing is expected to be more efficient.

As illustrated in Figure 1, during the query phase, we construct the super-network with adjacency matrix $\hat{\boldsymbol{A}} = \boldsymbol{A}_1 || \boldsymbol{A}_2 || \ldots || \boldsymbol{A}_k$, and feature matrix $\hat{\boldsymbol{X}} = \boldsymbol{X}_1 || \boldsymbol{X}_2 || \ldots || \boldsymbol{X}_k$, where $||$ denotes the logical OR operation. The super-network $(\hat{\boldsymbol{A}}, \hat{\boldsymbol{X}})$ is then trained by uniformly sampling from the architectures $\{(\boldsymbol{A}_i, \boldsymbol{X}_i)\}$ [5]. In each iteration, one sub-network $(\boldsymbol{A}_i, \boldsymbol{X}_i)$ is randomly sampled from the super-network, and only the corresponding (forward and backward propagation) paths in it are activated. Finally, we evaluate each sub-network by only forwarding data along the corresponding paths in the super-network.

In ENAS [24], the operation weights are reused along the whole search process. Hence, networks evaluated later in the process are trained with longer budgets, which may render the evaluation unfair. In the proposed algorithm, we reinitialize the operation weights at each query phase, ensuring that each sub-network is trained for the same number of iterations.

## 3.3 Algorithm BONAS

The whole procedure, which will be called BONAS (Bayesian Optimized Neural Architecture Search), is shown in Algorithm 2. Given the search space $\mathcal{A}$, we start with a set $\mathcal{D}$ of $m_0$ random architectures $\{(\boldsymbol{A}_i, \boldsymbol{X}_i)\}$, which have been queried and the corresponding performance values $\{t_i\}$ known. The GCN embedding extractor and BSR are then trained using $\mathcal{D}$ (Section 3.1).

In each search iteration, a pool $\mathcal{C}$ of candidates are sampled from $\mathcal{A}$ by evolutionary algorithm (EA). For each candidate, its embedding is generated by the GCN, which is used by BSR to compute the mean and variance of its predicted accuracy. The UCB score is then obtained from (1). Candidates with the top-$k$ UCB scores are selected and queried using weight sharing (Section 3.2). The evaluated models and their performance values are added to $\mathcal{D}$. The GCN predictor and BSR are then updated using the enlarged $\mathcal{D}$. The procedure is repeated until convergence.

## 4 Experiments

In the following experiments, we use NAS-Bench-101 [38], which is the largest NAS benchmark data set (with 423K convolutional architectures), and the more recent NAS-Bench-201 [8], which uses a different search space (with 15K architectures) and is applicable to almost any NAS algorithm.

As both NAS-Bench-101 and NAS-Bench-201 focus on convolutional architectures, we also construct another benchmark data set (denoted LSTM-12K), containing 12K LSTM models trained on the Penn TreeBank data set [21] following the same setting in ENAS [24]. Each LSTM cell, with an adjacency matrix and a list of operations, is represented by a string. In the search space, there are 4 possible activation functions (tanh, ReLU, identity, and sigmoid) and each node takes one previous node as input. The architecture is obtained by selecting the activation functions and node connections. Due to limitation on computational resources, we only sample architectures with 8 or fewer nodes. We

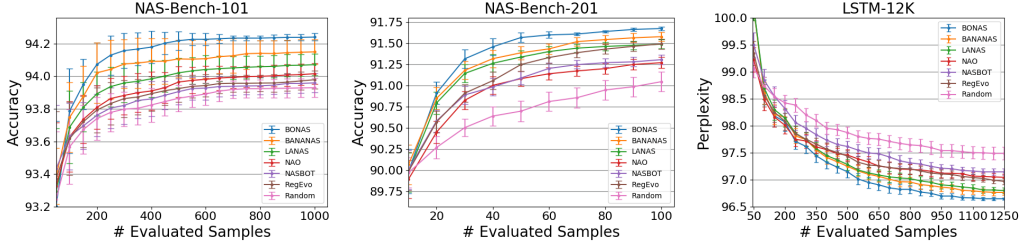

Figure 4: Performance (accuracy/perplexity) of the best model versus number of evaluated samples.

randomly sampled 12K cell structures. The perplexity is used as the metric to evaluate performances of the models. More details on the training setup and search space are in Appendix B.

Experiments are also performed in the open-domain scenario using the NASNet search space [42]. All experiments are performed on NVIDIA Tesla V100 GPUs.

## 4.1 Comparison of Predictor Performance

In this section, we demonstrate superiority of the proposed GCN predictor over existing MLP and LSTM predictors in [34], and the meta NN in [35]. The GCN has four hidden layers with 64 units each. Training is performed by minimizing the square loss, using the Adam optimizer [12] with a learning rate of 0.001 and a mini-batch size of 128. For the MLP predictor, we follow [34] and use 5 fully-connected layers, with 512, 2048, 2048, 512 and 1 units, respectively. As for the LSTM predictor, the sizes of both the hidden layer and embedding are 100. The last LSTM hidden layer is connected to a fully-connected layer. For the meta NN, we use an ensemble of 3 predictors (each being a fully-connected neural network with 10 layers, and 20 hidden units in each layer) and apply the full-path encoding scheme.

Experiments are performed on the NAS-Bench-101, NAS-Bench-201, and LSTM-12K data sets. For each data set, we use 85% of the data for training, 10% for validation, and the rest for testing. For performance evaluation, as in [34], we use the correlation coefficient between the model's predicted and actual performance values (i.e., testing accuracy on NAS-Bench-101 and NAS-Bench-201, and perplexity on LSTM-12K). Table 1 shows the results. As can be seen, the GCN predicts the performance more accurately than the other three predictors.

Table 1: Correlation between the model's predicted and actual performance.

|  | NAS-Bench-101 | NAS-Bench-201 | LSTM-12K |
|---|---|---|---|
| MLP | 0.830 | 0.865 | 0.530 |
| LSTM | 0.741 | 0.795 | 0.560 |
| Meta NN | 0.648 | 0.967 | 0.582 |
| GCN | **0.841** | **0.973** | **0.742** |

## 4.2 Closed Domain Search

In this section, we study the search efficiency of the proposed BONAS algorithm. In step 1 of Algorithm 2, we start with 10 random architectures that are fully trained.[3] In step 4, for NAS-Bench-201, the adjacency matrix is fixed and not mutated by the EA sampler. The candidate pool $\mathcal{C}$ is of size 10,000 and $\gamma = 0.5$ in (1). As the benchmarks already contain the performances of all models, we query each architecture ($k = 1$) by directly obtaining its accuracy from the data sets rather than using the weight-sharing query scheme in Section 3.2. Since only the search phase but not the query phase is performed, this allows us to demonstrate the search efficiency of BONAS more clearly.

Note that one-shot methods do not search for models iteratively. We compare BONAS with the following state-of-the-art sample-based NAS baselines: (i) Random search [37], which explores the search space randomly without exploitation; (ii) Regularized evolution [25], which uses a heuristic evolution process for exploitation; (iii) NASBOT [11], which uses BO with a manually-defined kernel on architectures; (iv) Neural Architecture Optimization (NAO) [19], which finds the architecture in a continuous embedding space with gradient descent; (v) LaNAS [33], which estimates the architecture

Table 2: Performance of open-domain search on CIFAR-10. #blocks is the number of blocks in a cell, and cutout [7] is a popular data augmentation strategy in NAS. For one-shot NAS methods, the samples are not explored one by one, and the number of samples evaluated is marked "-".

| | #blocks | #params | top-1 err (%) | #samples evaluated | GPU days |
|---|---|---|---|---|---|
| GHN+cutout [39] | 7 | 5.7 M | 2.84 | - | 0.84 |
| LaNet+cutout [33] | 7 | 3.2 M | 2.53 | 803 | 150 |
| ASNG-NAS+cutout [1] | 5 | 3.9 M | 2.83 | - | 0.11 |
| ENAS+cutout [24] | 5 | 4.6 M | 2.89 | - | 0.45 |
| NASNet-A+cutout [42] | 5 | 3.3 M | 2.65 | 20,000 | 2,000 |
| AmoebaNet-B+cutout [25] | 5 | 2.8 M | 2.55 | 27,000 | 3,150 |
| NAO [19] | 5 | 10.6 M | 3.18 | 1,000 | 200 |
| DARTS+cutout [18] | 4 | 3.3 M | 2.76 | - | 1.5 |
| BayesNAS+cutout [40] | 4 | 3.4 M | 2.81 | - | 0.2 |
| PNASNet-5 [17] | 4 | 3.2 M | 3.41 | 1,160 | 225 |
| BANANAS+cutout [35] | 4 | 3.6 M | 2.64 | 100 | 11.8 |
| BONAS-A+cutout | 4 | 3.45 M | 2.69 | 1,200 | 2.5 |
| BONAS-B+cutout | 4 | 3.06 M | 2.54 | 2,400 | 5.0 |
| BONAS-C+cutout | 4 | 3.48 M | 2.46 | 3,600 | 7.5 |
| BONAS-D+cutout | 4 | 3.30 M | **2.43** | 4,800 | 10.0 |

performance in a coarse model subspace by Monte Carlo tree search and (vi) BANANAS [35], which applies a traditional BO framework for the NAS problem. The experiment is repeated 50 times, and the averaged result with standard deviation reported.

Following [33, 34], Figure 4 shows the performance of the best model after using a given number of architecture samples. As can be seen, BONAS consistently outperforms the other search algorithms.

## 4.3 Open Domain Search

In this section, we perform NAS on the NASNet search space [42] using the CIFAR-10 data set. Following [18], we allow 4 blocks inside a cell. In step 10 of Algorithm 2, $k = 100$ models are merged to a super-network and trained for 100 epochs using the procedure discussed in Section 3.2. In each epoch, every sub-network is trained for the same number of iterations. The other experimental settings are the same as in Section 4.2.

The proposed BONAS algorithm is compared with the following state-of-the-art sample-based NAS algorithms (results of the various baselines are taken from the corresponding papers): (i) NASNet [42], which uses reinforcement learning to sample architectures directly; (ii) AmoebaNet [25], which finds the architecture by an evolution algorithm; (iii) PNASNet [17], which searches the architecture progressively combined with a predictor; (iv) NAO [19]; (v) LaNet [33]; (vi) BANANAS. We also show results of the one-shot NAS methods, including: (i) ENAS [24], which finds the model by parameter sharing; (ii) DARTS [18], which applies continuous relaxation for super-network training; (iii) BayesNAS [40], which considers the dependency on architectures; and (iv) ASNG-NAS [1], which proposes a stochastic natural gradient method for the NAS problem.

Results on CIFAR10 are shown in Table 2. We list 4 BONAS models (A, B, C, D) obtained with different numbers of evaluated samples. Note that different papers may use different numbers of blocks in the experiment, and comparison across different search spaces may not be fair. As can be seen from Table 2, BONAS outperforms all the other algorithms in terms of the top-1 error. Moreover, by using weight-sharing query, BONAS is very efficient compared with the other sample-based NAS algorithms. For example, BONAS can sample and query 4800 models in around 10 GPU days, while BANANAS only queries 100 models in 11.8 GPU days. We show the search progress of BONAS in Appendix C, and example architectures learned by BONAS are in Appendix D.

## 4.4 Transfer Learning

As in [18, 19, 33], we consider transferring the architectures learned from CIFAR-10 to ImageNet [6]. We follow the mobile setting in [18, 42]. The size of input image is $224 \times 224$ and the number of multiply-add operations is constrained to be fewer than 600M. Other training setups are the same

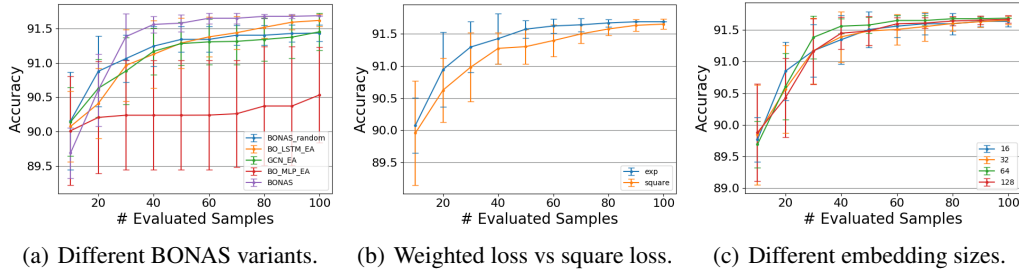

| (a) Different BONAS variants. | (b) Weighted loss vs square loss. | (c) Different embedding sizes. |

Figure 5: Ablation study on NAS-Bench-201.

as in [18]. Since BONAS-A is not competitive on CIFAR-10 compared with other baselines in Section 4.3, we only fully train BONAS-B/C/D on ImageNet.

Results are shown in Table 3 (results of the baselines are from the corresponding papers). As can be seen, in the search space with 4 blocks (as used by BONAS), the transferred architecture found by BONAS outperforms the others. It's remarkable that BONAS-C achieves a top-1 error of $24.6\%$ and a top-5 error of $7.5\%$ on ImageNet. This transferred architecture remains competitive even when compared with baselines using different numbers of blocks in the search space.

Table 3: Transferability of different learned architectures on ImageNet. Here, #blocks is the number of blocks inside the cell, and #mult-adds is the number of multiply-add operations.

|  | #blocks | #mult-adds | #params | error (%) top-1 | error (%) top-5 |
|---|---|---|---|---|---|
| LaNet | 7 | 570 M | 5.1 M | 25.0 | 7.7 |
| NASNet-A | 5 | 564 M | 5.3 M | 26.0 | 8.4 |
| NASNet-B | 5 | 488 M | 5.3 M | 27.2 | 8.7 |
| NASNet-C | 5 | 558 M | 4.9 M | 27.5 | 9.0 |
| AmoebaNet-A | 5 | 555 M | 5.1 M | 25.5 | 8.0 |
| AmoebaNet-B | 5 | 555 M | 5.3 M | 26.0 | 8.5 |
| AmoebaNet-C | 5 | 570 M | 6.4 M | 24.3 | 7.6 |
| PNASNet-5 | 4 | 588 M | 5.1 M | 25.8 | 8.1 |
| DARTS | 4 | 574 M | 4.7 M | 26.7 | 8.7 |
| BayesNAS | 4 | 440 M | 4.0 M | 26.5 | 8.9 |
| BONAS-B | 4 | 500 M | 4.5 M | 24.8 | 7.7 |
| BONAS-C | 4 | 557 M | 5.1 M | 24.6 | 7.5 |
| BONAS-D | 4 | 532 M | 4.8 M | 25.4 | 8.0 |

## 4.5 Ablation Study

In this section, we perform ablation study on NAS-Bench-201. The experiment settings are the same as in Section 4.2. To investigate the effect of different components of the proposed model, we study the following BONAS variants: (i) BONAS_random, which replaces EA sampling with random sampling; (ii) BO_LSTM_EA, which replaces the GCN predictor by LSTM; (iii) BO_MLP_EA, which replaces the GCN predictor by MLP; (iv) GCN_EA, which removes Bayesian sigmoid regression and uses the GCN output directly as selection score. Results are shown in Figure 5(a). As can be seen, BONAS outperforms the various variants.

Next, we compare the proposed weighted loss in (2) with traditional square loss. Experimental results on NAS-Bench-201 are shown in Figure 5(b). As can be seen, the use of the proposed loss improves search efficiency by paying different emphasis on different models.

Finally, to verify the robustness of the BONAS model, we also investigate the influence of embedding size ($\{16, 32, 64, 128\}$). As can be seen from Figure 5(c), the performance is robust to the GCN embedding size.

## 5 Conclusion

In this paper, we proposed BONAS, a sample-based NAS method combined with weight-sharing paradigm for use with BO. In the search phase, we use GCN with the Bayesian sigmoid regressor as BO's surrogate model to search for top-performing candidate architectures. As for query, we adapt the weight-sharing mechanism to query a batch of promising candidate architectures together. BONAS accelerates sample-based NAS methods and has robust results, thus bridging the gap between sample-based and one-shot NAS methods. Experiments on closed-domain search demonstrate its efficiency compared with other sample-based algorithms. As for open-domain search, we validate BONAS in the NASNet search space, and the obtained architecture achieves a top-1 error of $2.43\%$ on CIFAR-10 in 10 GPU days.

## Broader Impact

Neural Architecture Search (NAS) is a powerful framework, and widely used in the industry to automatically search for models with good performance. However, the large number of architecture samples required and the consequent heavy computation are key obstacles for many researchers and small businesses. NAS also introduces environmental issues that cannot be overlooked. As pointed out in [32], the $CO_2$ emission from a NAS process can be comparable to that from 5 cars' lifetime. With the proposed approach, the above-mentioned issues can be alleviated without compromising the final model's performance.

BONAS provides insights to future NAS research and industrial applications. It allows researchers and businesses with limited compute to conduct NAS experiments. This new NAS algorithm is also expected to be more energy-efficient and environmentally friendly.

## Footnotes

[1]The code is available at https://github.com/pipilurj/BONAS.

[2]The logit-normal distribution is given by: $p(t; \mu, \sigma) = \frac{1}{\sigma\sqrt{2\pi}} \frac{1}{t(1-t)} \exp\left( -\frac{(\text{logit}(t) - \mu)^2}{2\sigma^2} \right)$.

[3]These 10 are counted towards the total number of architectures sampled.

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
