[Supplementary Material · Appendix.pdf]

# Appendix A   Proofs

**Proposition 2.** *Let $\mathcal{N}(x|\mu, \sigma^2)$ be the normal distribution with mean $\mu$ and variance $\sigma^2$. For any given $\alpha$ and $\beta$,*

$$\int \Phi(\alpha(x+\beta))\mathcal{N}(x|\mu,\sigma^2)dx = \Phi\left(\frac{\alpha(\mu+\beta)}{(1+\alpha^2\sigma^2)^{1/2}}\right), \tag{5}$$

*where $\Phi(x) = \int_{-\infty}^{x}\mathcal{N}(z|0,1)dz$ is the cumulative distribution function of the standard normal distribution.*

*Proof.* Let $z = (x-\mu)/\sigma$, we have

$$
\begin{aligned}
y(\mu,\sigma) &= \int \Phi(\alpha(x+\beta))\mathcal{N}(x|\mu,\sigma^2)dx \\
&= \int \Phi(\alpha(\mu+\sigma z+\beta))\frac{1}{(2\pi\sigma^2)^{1/2}}\exp\{-\frac{1}{2}z^2\}\sigma dz \\
&= \int \Phi(\alpha(\mu+\sigma z+\beta))\frac{1}{(2\pi)^{1/2}}\exp\{-\frac{1}{2}z^2\}dz.
\end{aligned}
$$

Take the derivative of $y$ with respect to $\mu$,

$$
\begin{aligned}
\frac{\partial y(\mu,\sigma)}{\partial \mu} &= \frac{\alpha}{2\pi}\int \exp\{-\frac{1}{2}z^2 - \frac{1}{2}\alpha^2(\mu+\sigma z+\beta)^2\}dz \\
&= \frac{\alpha}{2\pi}\int \exp\{-\frac{1}{2}z^2 - \frac{1}{2}\alpha^2(\mu^2+\sigma^2 z^2+\beta^2+2\mu\sigma z+2\mu\beta+2\sigma z\beta)\}dz \\
&= \frac{\alpha}{2\pi}\int \exp\{-\frac{1}{2}(1+\alpha^2\sigma^2)(z^2+\frac{2\alpha^2\sigma(\mu+\beta)}{1+\alpha^2\sigma^2}z+\frac{\alpha^2(\mu^2+\beta^2+2\mu\beta)}{1+\alpha^2\sigma^2})\}dz \\
&= \frac{\alpha}{2\pi}\int \exp\{-\frac{1}{2}(1+\alpha^2\sigma^2)((z+\frac{\alpha^2\sigma(\mu+\beta)}{1+\alpha^2\sigma^2})^2 - \frac{\alpha^4\sigma^2(\mu+\beta)^2}{(1+\alpha^2\sigma^2)^2}+\frac{\alpha^2(\mu+\beta)^2}{1+\alpha^2\sigma^2})\}dz \\
&= \frac{\alpha}{2\pi}\int \exp\{-\frac{1}{2}(1+\alpha^2\sigma^2)(z+\frac{\alpha^2\sigma(\mu+\beta)}{1+\alpha^2\sigma^2})^2 + \frac{1}{2}\frac{\alpha^4\sigma^2(\mu+\beta)^2}{1+\alpha^2\sigma^2} - \frac{1}{2}\alpha^2(\mu+\beta)^2\}dz \\
&= \frac{\alpha}{2\pi}\int \exp\{-\frac{1}{2}(1+\alpha^2\sigma^2)(z+\frac{\alpha^2\sigma(\mu+\beta)}{1+\alpha^2\sigma^2})^2 - \frac{1}{2}\frac{\alpha^2(\mu+\beta)^2}{1+\alpha^2\sigma^2}\}dz \\
&= \frac{\alpha}{2\pi}\exp\{-\frac{1}{2}\frac{\alpha^2(\mu+\beta)^2}{1+\alpha^2\sigma^2}\}\int \exp\{-\frac{1}{2}(1+\alpha^2\sigma^2)(z+\frac{\alpha^2\sigma(\mu+\beta)}{1+\alpha^2\sigma^2})^2\}dz \\
&= \frac{1}{(2\pi)^{1/2}}\frac{\alpha}{(1+\alpha^2\sigma^2)^{1/2}}\exp\{-\frac{1}{2}\frac{\alpha^2(\mu+\beta)^2}{1+\alpha^2\sigma^2}\}.
\end{aligned}
$$

Similarly, take the derivative of $y$ with respect to $\sigma$,

$$
\begin{aligned}
\frac{\partial y(\mu,\sigma)}{\partial \sigma} &= \frac{\alpha}{2\pi}\int \exp\{-\frac{1}{2}z^2 - \frac{1}{2}\alpha^2(\mu+\sigma z+\beta)^2\}zdz \\
&= \frac{\alpha}{2\pi}\exp\{-\frac{1}{2}\frac{\alpha^2(\mu+\beta)^2}{1+\alpha^2\sigma^2}\}\int \exp\{-\frac{1}{2}(1+\alpha^2\sigma^2)(z+\frac{\alpha^2\sigma(\mu+\beta)}{1+\alpha^2\sigma^2})^2\}zdz \\
&= -\frac{1}{(2\pi)^{1/2}}\frac{\alpha^3\sigma(\mu+\beta)}{(1+\alpha^2\sigma^2)^{3/2}}\exp\{-\frac{1}{2}\frac{\alpha^2(\mu+\beta)^2}{1+\alpha^2\sigma^2}\}.
\end{aligned}
$$

Note that

$$
\begin{aligned}
\frac{\partial \Phi(\frac{\alpha(\mu+\beta)}{(1+\alpha^2\sigma^2)^{1/2}})}{\partial \mu} &= \frac{1}{(2\pi)^{1/2}}\frac{\alpha}{(1+\alpha^2\sigma^2)^{1/2}}\exp\{-\frac{1}{2}\frac{\alpha^2(\mu+\beta)^2}{1+\alpha^2\sigma^2}\}, \\
\frac{\partial \Phi(\frac{\alpha(\mu+\beta)}{(1+\alpha^2\sigma^2)^{1/2}})}{\partial \sigma} &= -\frac{1}{(2\pi)^{1/2}}\frac{\alpha^3\sigma(\mu+\beta)}{(1+\alpha^2\sigma^2)^{3/2}}\exp\{-\frac{1}{2}\frac{\alpha^2(\mu+\beta)^2}{1+\alpha^2\sigma^2}\}.
\end{aligned}
$$

Thus,

$$\int \Phi(\alpha(x+\beta))\mathcal{N}(x|\mu,\sigma^2)dx = \Phi\left(\frac{\alpha(\mu+\beta)}{(1+\alpha^2\sigma^2)^{1/2}}\right) + C,$$

for some constant $C$. When $\alpha = 0$,

$$y(\mu,\sigma) = \int \Phi(0)\mathcal{N}(z|0,1)dz = \frac{1}{2} = \Phi(0),$$

where Eq. (5) always holds. When $\alpha \neq 0$, consider the case where $\mu = -\beta, \sigma = \frac{1}{\alpha}$,

$$
\begin{aligned}
y(-\beta, \frac{1}{\alpha}) &= \int \Phi(z)\mathcal{N}(z|0,1)dz \\
&= \int (\Phi(z) - \frac{1}{2})\mathcal{N}(z|0,1)dz + \int \frac{1}{2}\mathcal{N}(z|0,1)dz \\
&= \int \frac{1}{2}\mathcal{N}(z|0,1)dz \\
&= \frac{1}{2} = \Phi(\frac{\alpha(\mu+\beta)}{(1+\alpha^2\sigma^2)^{1/2}}|_{\mu=-\beta,\sigma=\frac{1}{\alpha}}),
\end{aligned}
$$

which means $C = 0$.

$$\implies \int \Phi(\alpha(x+\beta))\mathcal{N}(x|\mu,\sigma^2)dx = \Phi(\frac{\alpha(\mu+\beta)}{(1+\alpha^2\sigma^2)^{1/2}}).$$

$\square$

In the following, we align the function $\text{sigmoid}(x)$ with $\Phi(\lambda x)$ (where $\Phi$ is as defined in Proposition 1) such that $\text{sigmoid}(x) \approx \Phi(\lambda x)$. Obviously, these two functions have the same maxima, minima, and center (at $x = 0$). Thus, we only need to align their derivatives at $x = 0$. Now,

$$
\begin{aligned}
\frac{\partial \text{sigmoid}(x)}{\partial x}|_{x=0} &= e^{-x}(1+e^{-x})^{-2}|_{x=0} = \frac{1}{4}, \\
\frac{\partial \Phi(\lambda x)}{\partial x}|_{x=0} &= \frac{\lambda}{(2\pi)^{1/2}} \exp\{-\frac{1}{2}(\lambda x)^2\}|_{x=0} = \frac{\lambda}{(2\pi)^{1/2}}.
\end{aligned}
$$

This implies $\lambda^2 = \frac{\pi}{8}$.

Similarly, we also align $(\text{sigmoid}(x))^2$ with $\Phi(\lambda\alpha(x+\beta))$, for some appropriate $\alpha$ and $\beta$. Again, note that both functions have the same maxima and minima. The center of $\Phi(\lambda\alpha(x+\beta))$ is at $(-\beta, 1/2)$. For alignment, we consider the point when $(\text{sigmoid}(x))^2 = 1/2$ as its center point, where $x = \log(\sqrt{2}+1)$. It is easy to see that $\beta = -\log(\sqrt{2}+1)$. As for the derivative at this center,

$$
\begin{aligned}
\frac{\partial (\text{sigmoid}(x))^2}{\partial x}|_{x=-\beta} &= 2e^{-x}(1+e^{-x})^{-3}|_{x=-\beta} = (2-\sqrt{2})/2, \\
\frac{\partial \Phi(\lambda\alpha(x+\beta))}{\partial x}|_{x=-\beta} &= \frac{\lambda\alpha}{(2\pi)^{1/2}} \exp\{-\frac{1}{2}(\lambda\alpha(x+\beta))^2\}|_{x=-\beta} = \frac{\lambda\alpha}{(2\pi)^{1/2}},
\end{aligned}
$$

which implies $\alpha = 4 - 2\sqrt{2}$. Illustrations of the approximations are shown in Figure 6.

(a) $\text{sigmoid}(x)$.      (b) $(\text{sigmoid}(x))^2$.

Figure 6: Approximations of $\text{sigmoid}(x)$ and $(\text{sigmoid}(x))^2$.

Now, using Proposition 1 and the above approximations, we have

$$
\begin{aligned}
E[t] &= \int \frac{1}{\sigma\sqrt{2\pi}} \frac{1}{1-t} \exp(-\frac{(logit(t)-\mu)}{2\sigma^2})dt \\
&= \int \mathrm{sigmoid}(x)\mathcal{N}(x|\mu,\sigma^2)dx \simeq \int \Phi(\lambda x)\mathcal{N}(x|\mu,\sigma^2)dx \\
&= \Phi(\frac{\lambda\mu}{\sqrt{1+\lambda^2\sigma^2}}) \\
&\simeq \mathrm{sigmoid}(\frac{\mu}{\sqrt{1+\lambda^2\sigma^2}}), \\
E[t^2] &= \int \frac{1}{\sigma\sqrt{2\pi}} \frac{t}{1-t} \exp(-\frac{(logit(t)-\mu)}{2\sigma^2})dt \\
&= \int \mathrm{sigmoid}(x)^2\mathcal{N}(x|\mu,\sigma^2)dx \\
&\simeq \int \Phi(\lambda\alpha(x+\beta))\mathcal{N}(x|\mu,\sigma^2)dx \\
&= \Phi(\frac{\lambda\alpha(\mu+\beta)}{\sqrt{1+\lambda^2\alpha^2\sigma^2}}) \\
&\simeq \mathrm{sigmoid}(\frac{\alpha(\mu+\beta)}{\sqrt{1+\lambda^2\alpha^2\sigma^2}}), \\
var[t] &= E[t^2] - E[t]^2 \\
&\simeq \mathrm{sigmoid}(\frac{\alpha(\mu+\beta)}{\sqrt{1+\lambda^2\alpha^2\sigma^2}}) - (\mathrm{sigmoid}(\frac{\mu}{\sqrt{1+\lambda^2\sigma^2}}))^2
\end{aligned}
$$

## Appendix B   Data set

### B.1   LSTM-12K Data Set

We randomly sampled 12K cell structures from the same search space as used in [26]. The data set consists of 9000 architectures with 7-node cells and 3000 architectures with 8-node cells. There are 4 choices of operations: ReLU, Sigmoid, Tanh, Identity. Each architecture is trained for 10 epochs on the PTB data set [23]. Other training setups are the same as [26]. Specifically, we use SGD with a learning rate of $20.0$ to train our LSTM models and clip the norm of the gradient at $0.25$. Besides, we also adapt three same regularization techniques: (i) an $\ell_2$-regularizer with weight decay parameter $10^{-7}$; (ii) dropout [9] with a rate of $0.4$; (iii) tying of the word embeddings and softmax weights [11]. The models' cell structures, numbers of parameters and perplexities are recorded. This data set can be used to test the efficiency of NAS algorithms before applying them in the open domain.

### B.2   NASNet Search Space

We follow the search space setting of DARTS [20], in which the architecture is obtained by stacking the learned cell. Each cell consists of $4$ blocks, two inputs (outputs of the previous cell and previous previous cell), and one output. Each intermediate block contains two inputs and one output as follows:

$$x^{(i)} = o^{(i,j)}(x^j) + o^{(i,k)}(x^k),$$

where $x^{(i)}$ is the block output, and $x^{(j)}, x^{(k)}$ are any two predecessors. There are 7 types of allowed operations: $3 \times 3$ and $5 \times 5$ separable convolutions, $3 \times 3$ and $5 \times 5$ dilated separable convolutions, $3 \times 3$ max pooling, $3 \times 3$ average pooling and identity.

Similar to [19], we apply the same cell architecture for both "normal" and "reduction" layers. In the proposed GCN predictor, each operation is treated as a node, and each data flow as an edge.

To train the architecture, we use the same setting as in [20]. We use momentum SGD (with learning rate $0.025$ (anneal cosine strategy), momentum $0.9$, and weight decay $3 \times 10^{-4}$).

# Appendix C    Illustration of Efficient Estimation

To demonstrate efficiency of the proposed estimation scheme using weight-sharing, Figure 7 shows the search progress of BONAS on the open domain search in Section 4.3. Each point in the figure represents a selected architecture. For each given number of samples searched, a Gaussian kernel density estimator is fitted on the accuracy distribution of the selected architectures. The color corresponds to the corresponding probability density function value. As can be seen, when very few architectures are searched, the surrogate model cannot estimate the architecture accuracy well, and the accuracy distribution of the selected models is diffuse. With more and more samples, the GCN and BSR can perform the accuracy estimation better. After around 2000 samples, most of the candidate models selected by BONAS have high estimated accuracies.

For sub-networks that are sampled in a particular search iteration, Figure 8 compares their actual accuracies (obtained by full training) with the estimated accuracies obtained by the proposed method (Section 3.2) and standard weight-sharing (which constructs the super-network by using all models in the search space). As can be seen, the proposed weight-sharing among a smaller number of promising models can achieve higher correlation.

Figure 7: Visualization of BONAS's search progress on open-domain search.

Figure 8: Actual accuracy versus estimated accuracies obtained by the proposed method and one-shot NAS.

# Appendix D    Example Architectures Obtained

Figure 9 shows some example architectures that are obtained by the proposed method from open-domain search on the NASNet search space (Section 4.3).

(a) BONAS-A.

(b) BONAS-B.

(c) BONAS-C.

(d) BONAS-D.

Figure 9: Example models obtained by BONAS in the NASNet search space.