[Reviews · NeurIPS 2020]

Review 1

Summary and Contributions: This paper proposes a novel sample-based Bayesian optimized NAS algorithm - BONAS. Different from previous GP-based BO, they introduce a GCN embedding extractor and Bayesian Sigmoid Regressor as the surrogate model and some theoretical analysis is provided. They incorporate weight sharing mechanism to accelerate the evaluation phase. The experiment results demonstrate that BONAS outperforms other baselines on different tasks.

Strengths: 1. GCN embedding extractor is novel for NAS and their encoding scheme is direct since architecture can be represented by a DAG. Table 1 shows the performance of GCN compared with other predictors (MLP, LSTM). 2. Bayesian Sigmoid Regressor, a variant of traditional Bayesian Linear Regreeor, is proposed for BO’s surrogate model. Theoretical analysis is also provided and Figure 3 show their approximation results. 3. The author split sample-based NAS framework into two parts: search phase and query phase. Most previous methods only consider search phase and BONAS focus on query phase both with the help of weight sharing paradigm. 4. The authors present a new dataset LSTM-12K and it’s very valuable for NAS community. 5. Since NAS is an application domain with much attention, the result is important and their experiment results are very good, both on close domain and open domain (Figure 4 and Table 2).

Weaknesses: 1. In Table 3, what’s the meaning of the top-1 error? Best accuracy or average accuracy? Error bar can be showed in Table 3 for better comparison. 2. Do you compare with some HPO methods such as BOHB in the closed domain search. 3. Will the code and the LSTM-12K be open-sourced? 4. Better highlight the best results in Table 3 for a clear representation. 5. The final results in Table 2 is not fair. The result of adding auto-augment for your method should be removed.

Correctness: Yes

Clarity: Yes

Relation to Prior Work: Yes

Reproducibility: Yes

Additional Feedback:


Review 2

Summary and Contributions: This paper proposes a method of neural architecture search (NAS) using the Bayesian optimization-based surrogate model for predicting architecture performance. The proposed Bayesian optimized neural architecture search (BONAS) uses a graph convolutional network as a surrogate model and trains the promising architectures by one-shot NAS manner through weight sharing. BONAS can reduce the computational cost of sample-based NAS due to one-shot training. The effectiveness of BONAS is evaluated on the numerical experiments on image classification tasks. # Comments after the author response Thank you for answering my questions. The authors clearly answer my questions, and my concern has been solved. I raise my score to 7.

Strengths: Although several neural architecture search methods using a surrogate model (performance predictor) have been proposed so far, this paper incorporates the one-shot architecture training into the surrogate model-based NAS. The combination of one-shot architecture training and surrogate model-based NAS is interesting.

Weaknesses: BONAS introduces the additional hyper-parameters, such as the architecture of GCN and training hyper-parameters of GCN, into the NAS system. The robustness against such a hyper-parameter setting should be investigated.

Correctness: The claims and method in this paper are probably correct. The claims are supported by the experimental evaluation.

Clarity: The writing quality is acceptable.

Relation to Prior Work: The discussion about the related work is enough.

Reproducibility: Yes

Additional Feedback: How sensitive is the performance of BONAS against the hyper-parameter setting of GCN? In the experiment on CIFAR-10, the impacts of the one-shot training and the use of GCN should be investigated. Namely, it is nice if BONAS with LSTM- or MLP-based surrogate model is compared to the proposed BONAS to show the effectiveness of GCN as a surrogate model.


Review 3

Summary and Contributions: This paper use GCN as an architecture representation learner and use Bayesian Regressor as a neural predictor to surrogate the model performance. To speed up the NAS process, the network acquisitions are trained under the weight-sharing mechanism.

Strengths: The paper gives some key insight in NAS: 1). how architecture representation is learned matters in NAS, where they use GCN to learn the structure of neural nets in continuous space 2). Clustering neural architectures with similar performance in the nearby latent regions matters for neural predictor based methods. 3). adopting WS mechanism in sample-based NAS is useful to speed up NAS. Overall, I like the idea of this paper and proposed method is also technically reasonable.

Weaknesses: I have one major concern: the proposed method basically has two approximations. First, the architecture representation and search are jointly optimized given accuracy as the supervision signal, so it is not clear whether the discrete structure is well preserved in the continuous space in this way. Second, to speed up the search process the weight-sharing mechanism is adopted, however, it further couples the architecture representation learning and search, so I'm not sure whether it can guarantee the subnets accuracy trained with weight sharing correlates well with its final accuracy. A minor concern is which dimensionality you emb the architecture representation for Bayesian regression? In line 230 I see you use 512-d? As far as I know, BO typically works better in low-dimensional space (< 20-d).

Correctness: Line 15-16: "This is because weight-sharing among related architectures are more reliable than those in the one-shot approach." I don't understand what this means. Line 130: BANANAS's encoding sounds more like a computational-aware encoding rather than structure encoding? Other parts sounds technically good.

Clarity: It is well written. I appreciate the hard work of the authors.

Relation to Prior Work: Yes it is clearly discussed. This work and some related work like BANANAS open the new directions for NAS by studying the encoding (e.g. using GNN or path encoding) of neural networks. I would suggest authors take a look on some recent related work. [1] Graph Structure of Neural Networks. ICML 2020. [2] A Study on Encodings for Neural Architecture Search. arXiv:2007.04965 [3] Does Unsupervised Architecture Representation Learning Help Neural Architecture Search? arXiv:2006.06936 [4] FBNetV3: Joint Architecture-Recipe Search using Neural Acquisition Function. arXiv:2006.02049 [5] NASGEM: Neural Architecture Search via Graph Embedding Method. arXiv:2007.04452

Reproducibility: Yes

Additional Feedback: I would be appreciated if the authors can show sampled subnets trained with weight sharing has high correlation with trained with scratch using their method in DARTS search space. A proxy task is also fine if the relative ranking between subnets trained with WS and trained from scratch is well preserved. [Post rebuttal] I keep my original rating, though there are some open questions, the paper in its current form is already of interest to NAS community.


Review 4

Summary and Contributions: This paper addresses neural architecture search (NAS) with the following key contributions: 1) they mix sample-based and weight-sharing (one-shot) evaluation strategies into a hybrid strategy that allows them to evaluate more architectures at once; 2) they use a GCN based embedding method along with Bayestion Optimization to improve the efficiency of sampling new architectures to search over. One motivation is that weight-sharing to test all candidates at once is unreliable, so by instead evaluating a smaller subset of possible architectures the benefits of weight sharing accelerate feedback while the reduced number of candidates increases the quality of that feedback. They show in experiments that the GCN learns to score candidate architectures that better correlate with final performance. They report performance outperforming prior work across a number of benchmarks including NAS-Bench, CIFAR, and a newly constructed benchmark they provide for evaluating LSTMs.

Strengths: - the key ideas of this work are nontrivial and well motivated. For example, doing a hybrid evaluation strategy that samples and evaluates mulitple candidates at once with weight sharing is a good way to balance the trade-offs of the different existing evaluation approaches in NAS. - results across the board are strong on both NAS-Bench and CIFAR outperforming previous approaches - the evaluation in Table 1 is good to see comparing the correlation between their model's prediction and actual performance and demonstrating the effectiveness of their GCN design - I was going to comment on the lack of empirical justification for claims about the reliability of evaluation with weight sharing with a subset of the space versus the full space. It was good to see some results touching on this in the supplementary material (Figure 8). Though admittedly not sure that ~.6 vs a ~.7 correlation is really that big of a difference/the degree to which that would negatively impact your ability to find a top performing architecture. - addition of an LSTM benchmark to complement NAS-Bench - code is provided

Weaknesses: - main concern is that it is difficult to disentangle the contributions of each component to the final result. There are several contributions that make up the fully proposed system, but we don't have a clear picture of how important each one is. It seems very possible that one/some of the proposed details could be dropped for something simpler and not adversely affect performance. For example, if the architecture choice of the GCN is key in providing higher quality/higher correlation estimates of architecture performance how does the search go if the bayesian optimization is dropped entirely? What about the opposite, using BO without the GCN but an MLP/LSTM instead? Is the EA needed to produce the initial pool of candidates? While the results across a wide array of benchmarks are certainly compelling, the lack of any ablations along these lines is a bit of a bummer, especially since there are many facets of the system that are proposed and purportedly play a role in performance.

Correctness: As far as I can tell the claims/methodology is correct

Clarity: Yeah, overall the paper is written and organized reasonably well

Relation to Prior Work: For the most part, one point I was a little confused about was the comparison of weight sharing. My understanding of how weight sharing is used in ENAS is different from how it is used in this work. When a particular architecture is sampled in ENAS training is performed without resetting weights, hence there is sharing that takes place. But this is different from this paper that alternates sampling sub-networks over the course of training the super-network and evaluates them all at the end. I think this distinction could be made more clear to the reader.

Reproducibility: Yes

Additional Feedback: Overall I think this paper is strong enough to recommend acceptance, the ideas are interesting and well motivated and the evaluation across benchmarks is reasonably thorough. Misc questions: - for the GCN, were alternatives to a global node considered? For example, it is common to see pooling across all nodes used to get a final embedding - how was 100 decided upon for the number of candidates to test at once? It would be interesting to see how changing this number changes the sampling efficiency/quality/runtime of the search - were weights preserved across sampling rounds as in ENAS or reinitialized each time? the trade-off/reliabilty in weight sharing in this case seems like it would be a bit different than the impact of weight sharing when considering a simultaneous pool of candidates - is it possible to clarify the EA used to produced candidates, there wasn't too much discussion on why it was used and the degree to which it helped over randomly sampling candidates - the correlations reported in Table 1 are good, but seems like it would be useful to quantify the quality of the model's scoring estimates as the search progresses, that is, at initialization it is guiding the search having only seen a smaller pool of architectures, how good is the correlation at the beginnning and how does it improve over the course of the search? - would it be possible to report measures of variance for the open-domain search? If the search were run again from scratch, how consistent would it be? -- update post-rebuttal: the reviewers all seem to be on the same page that this is a good submission, I maintain my original recommendation for acceptance

[Author Response · NeurIPS 2020]



(a) **[R2, R4]** Ablation study.    (b) **[R2]** Embedding size.    (c) **[R3]** Visualization.    (d) **[R3, R4]** Weight sharing.

1. **[R1, R4]** *"meaning of top-1 error? Error bar ..."*, *"variance for the open-domain search"*: Top-1 error is simply the
2. classification error (a common terminology in image classification). The ImageNet experiment (Table 3) is expensive,
3. and so results reported are based on one single run (as in other papers). To show error bars, we use the open-domain
4. search experiment on CIFAR-10 (Table 2) instead, and re-train the architectures 5 times. The error bars (mean ± std)
5. are: BONAS-A $(2.69 \pm 0.05)$; BONAS-B $(2.54 \pm 0.04)$; BONAS-C $(2.46 \pm 0.03)$; BONAS-D $(2.43 \pm 0.03)$.
6. **[R1]** *"compare with some HPO methods such as BOHB in the closed domain search"*: BONAS outperforms BANANAS,
7. which in turn is better than BOHB (as shown in Figure 5.2 of BANANAS's arxiv version).
8. **[R1]** *"Will the code and the LSTM-12K be open-sourced?"*: Yes, as discussed in footnote 3.
9. **[R1]** *"The final results in Table 2 is not fair."*: Agree, we will remove it.

10. **[R2]** *"robustness against such a hyper-parameter setting"*: We use the same hyper-parameter setting for all 3 benchmarks.
11. Figure (b) above shows that the performance is robust to different GCN embedding sizes.
12. **[R2, R4]** *"nice if LSTM- or MLP-based surrogate model is compared"*, *"difficult to disentangle each component...clarify;*
13. *EA... degree to help"*: Figure (a) shows ablation study on NAS-Bench-201, which varies each component (surrogate
14. model/sampling method/BO) in the model. The other experimental settings are the same as in Section 4.2.

15. **[R3]** *"whether the discrete structure is well preserved in the continuous space"*: Figure (c) shows TSNE embedding of
16. the architecture embedding vectors. As can be seen, more accurate architectures are close to each other. Table 1 also
17. shows that one can obtain accurate prediction of the performance using the architecture embedding.
18. **[R3]** *"whether it can guarantee the subnets accuracy trained with weight sharing correlates well with its final accuracy*
19. *... appreciated if the authors can show sampled subnets trained with weight sharing has high correlation"*: We perform
20. an experiment similar to that reported in Figure 8 of Appendix, and compute the correlation between approximated
21. accuracy by weight-sharing and accuracy of fully-trained model. Instead of using 100 random models as in Figure 8,
22. we use 100 subnets that are trained with weight-sharing in the same round. Figure (d) above shows that the correlation
23. is high (0.832).
24. **[R3]** *"which dimensionality you emb the architecture? BO typically works better in low-dimensional..."*: We use
25. 64-d (line 227 in paper). In Table 7 of "Scalable Bayesian optimization using deep neural networks", it also uses
26. $\{50, 100, 200\}$ as embedding dimensionality.
27. **[R3]** *"weight-sharing among related architectures are more reliable than those in the one-shot approach"*: Standard
28. one-shot methods perform weight-sharing on a large set of subnets. These subnets can be very different and so sharing
29. their weights may not be a good assumption. Our method performs weight-sharing on a small subset of subnets, which
30. are similar in that they have high UCB scores (step 10 in Algorithm 2). Hence, sharing weights for this smaller subset
31. of similar-performance subnets may be more reasonable (as also verified by the high correlation between the actual and
32. approximate accuracies in Figure (d) mentioned above).
33. **[R3]** *"BANANAS's encoding sounds more like a computational-aware encoding ..."*: Agree, and we will clarify it.

34. **[R4]** *"not sure that .6 vs a .7 correlation is really that big of a difference"*: In Figure 8 of the Appendix, the models
35. are randomly sampled. Here, in Figure (d) above, we use subnets that are sampled in the same search iteration. The
36. correlation is higher (0.832), demonstrating that weight-sharing among a smaller subset of similar models is better.
37. **[R4]** *"a little confused about was the comparison of weight sharing..."*, *"were weights reinitialized each time?"*: Weights
38. are reinitialized each time across sampling rounds. Indeed, trained weights in ENAS is inherited along the whole search
39. process, but this approach may have some flaws: networks evaluated later are trained with longer budget than those
40. evaluated earlier, which may render the evaluation score unfair and cause misleading. Reinitializing the weights in each
41. sampling round can alleviate this issue since each sub-network is trained for the same iterations.
42. **[R4]** *"for the GCN, were alternatives to a global node considered? For example, it is common to see pooling"*: Yes, we
43. also tried differentiable pooling. The correlation is similar to adding a global node (0.840 vs 0.841 on NAS-Bench-101).
44. Thus, GCN propagation part is more important than how to add global node.
45. **[R4]** *"how was 100 decided upon for the number of candidates? how changing this number"*: This needs to be large
46. enough to accelerate search, but small enough to enable efficient weight sharing and exploration. We simply chose 100
47. as a convenient choice.
48. **[R4]** *"correlation at the beginning and how does it improve over the course of the search"*: The following are correlation
49. results over the course of search on NAS-Bench-201.

| # samples | 10 | 50 | 100 | 200 |
|---|---|---|---|---|
| correlation | 0.093 | 0.377 | 0.486 | 0.634 |

[Meta-Review · NeurIPS 2020]

Overall, the reviewers found the key ideas in the paper novel and well-motivated. I support the reviewers’ request for ablation studies to better disentangle the relative contribution of different components and the impact of different hyperparameters. Finally, please include standard deviations in table 2. They are readily available for the methods you are comparing against and the differences between methods are sufficiently small that it would be good to have an idea of variation across seeds.